# Evaluation of Clinical and Biochemical Traits in Egyptian Barki Sheep with Different Growth Performances

**DOI:** 10.3390/ani13060962

**Published:** 2023-03-07

**Authors:** Ragab M. Fereig, Rawia M. Ibrahim, Atef M. Khalil, Caroline F. Frey, Fatma A. Khalifa

**Affiliations:** 1Division of Internal Medicine, Department of Animal Medicine, Faculty of Veterinary Medicine, South Valley University, Qena 83523, Egypt; 2Division of Clinical and Laboratory Diagnosis, Department of Animal Medicine, Faculty of Veterinary Medicine, South Valley University, Qena 83523, Egypt; 3Division of Clinical Pathology, Department of Pathology and Clinical Pathology, Faculty of Veterinary Medicine, South Valley University, Qena 83523, Egypt; 4Institute of Parasitology, Department of Infectious Diseases and Pathobiology, Vetsuisse-Faculty, University of Bern, Länggassstrasse 122, CH-3012 Bern, Switzerland; 5Division of Infectious Diseases, Department of Animal Medicine, Faculty of Veterinary Medicine, South Valley University, Qena 83523, Egypt

**Keywords:** Barki sheep, parasite, mineral, biochemical, clinical, protein

## Abstract

**Simple Summary:**

Our randomized controlled clinical study revealed novel and valuable information on detecting biomarkers for growth performance in Barki sheep. Our study revealed that in most cases, the stunted Barki sheep exhibited a normal clinical picture, including appetite, and normal biochemical profiles for minerals and liver function markers. However, levels of nutritionally relevant biomarkers, such as total protein, albumin, cholesterol, and triglycerides, were markedly decreased in stunted sheep compared with normal sheep. These findings were also confirmed by lower levels of growth and thyroid hormones in stunted sheep than those in the control sheep. This approach is useful from economic and financial perspectives as it can be applied to exclude animals with poor growth and development from the livestock industry.

**Abstract:**

The Barki sheep industry is becoming increasingly important in Egypt because of the high quality of their meat and wool. This sheep breed is also commonly known for its resistance to arid and harsh environmental conditions. Such characteristics can be exploited in solving the problematic situation of inadequate animal protein for human consumption, particularly under climatic changes. However, very few studies have investigated aspects of breeding, nutrition, and susceptibility to infectious or non-infectious diseases in Barki sheep. Herein, we propose to unravel the differences in the clinical and biochemical profiles among Barki sheep of different growth rates. We measured clinical and biochemical parameters in stunted (*n* = 10; test group) and in good body condition (*n* = 9; control group) Barki sheep. Animals subjected to this experiment were of the same sex (female), age (12 months old), and housed in the same farm with similar conditions of feeding, management practice, and vaccination and deworming regimens. Regarding clinical examination, stunted/tested sheep showed a significantly higher pulse and respiratory rate compared to sheep with a good body condition/control group. The appetite, body temperature, and digestion processes were the same in both groups. In biochemical investigations, nutritional biomarkers were reduced markedly in stunted sheep compared with the control sheep, including total protein (*p* = 0.0445), albumin (*p* = 0.0087), cholesterol (*p* = 0.0007), and triglycerides (*p* = 0.0059). In addition, the Barki sheep test group suffered from higher levels of urea and blood urea nitrogen than the control group. Consistently, growth and thyroid hormone levels were lower in stunted sheep than the control sheep, although the differences were not statistically significant (*p* > 0.05). No significant differences were detected in both groups for serum levels of calcium, phosphorus, magnesium, iron, and zinc (*p* > 0.05). To detect the reasons for emaciation, certain debilitating infections were tested. All tested sheep showed negative coprological tests for gastrointestinal parasites, and had no obvious seropositivity to brucellosis, toxoplasmosis, neosporosis, or Q fever. This study demonstrates the useful biochemical markers for monitoring growth performance in Egyptian Barki sheep and unravels the usefulness of this breed in nationwide breeding and farming.

## 1. Introduction

Sheep are one of the primary sources of animal protein in human diets in Egypt and worldwide. Sheep can even endure in the desert by grazing on poor-quality fodder [1]. With 470,000 heads (11% of the total Egyptian sheep population), Barki sheep are raised in the Northwestern Coastal Desert of Egypt using a transhumant animal farming technique. Additionally, it is well known that this breed can survive in the arid conditions of Egypt [2]. Individual sheep of the same breed vary in terms of their final body weight and rate of growth. Thus, choosing lambs with rapid growth and a high final body weight is essential for enhancing meat production, which is the primary goal of sheep breeding projects [3].

Farm animal growth performance is a crucial economic feature that is influenced by both genetic and non-genetic variables, such as type of birth, sex, breed, season, age, and pre-mating weight of the dam [4,5].

The estimation of normal values of blood biochemical parameters is a highly significant tool for the laboratory interpretation of the clinical condition of tested sheep. Numerous factors, including age, sex, stress, weather, season, physical exercise, and pregnancy, can affect biochemical parameters. The degree of soundness and sickness of animals can be evaluated using biochemical parameters [6].

Growth hormone administration led to early puberty in Rahmani ewe lambs. This may be due to increased provision of trophic signals represented by increased gonadotropin-releasing hormone (GnRH) and luteinizing hormone (LH) secretions, which may arise from the early activation of the GnRH pulse generator. Furthermore, increased nutrient availability is crucial for ovarian function and/or local enhancement of ovarian activity. These effects may result from direct and/or indirect influences of somatotropin on the previously mentioned levels. Serum IGF-1 and/or blood-borne metabolites (mainly glucose) may be the potential signals by which somatotropin exerted its indirect effect [7].

The thyroid hormones, triiodothyronine (T3) and thyroxine (T4), relevant metabolic indices of animals’ nutritional state, maintain nutrient homeostasis, thermoregulation, growth, and productivity [8]. They are essential for normal growth and development of the fetus. T3 and T4 are required for the general accretion of fetal mass, elicit discrete developmental events in the fetal brain and somatic tissues from early gestation, and promote terminal differentiation of fetal tissues closer to term [9].

*Toxoplasma gondii* and *Neospora caninum* are intracellular protozoan parasites affecting sheep and can induce serious illness in lambs and severe reproductive disorders in pregnant ewes. In addition, *Brucella* spp. and *Coxiella burnetii* are bacterial agents with similar clinical forms of the above-mentioned protozoa. Previous reports document the high prevalence of the four pathogens among sheep in different regions of Egypt [10,11,12].

Despite the increasing significance of Barki sheep in different regions in Egypt, few studies on various aspects of biology and infection have been conducted on this breed. Herein, we measure clinical and biochemical parameters in stunted and good body condition Barki sheep. Animals used for this experiment were of the same sex (female), age (12 months old), and housed in the same farm with similar conditions of feeding, management practice, and vaccination and deworming regimens. A previous report had investigated some endocrine and biochemical parameters among fast- and slow-growing Muzaffarnagari male and female lambs during the postweaning period [13]. However, in the current study, we investigated sheep of a different breed (Barki), sex (female), location (Egypt), and demonstrated novel data on various examined clinical aspects, biochemical parameters, and metabolites. Furthermore, we provided more evidence for the health status of tested sheep using serological screening of a group of highly debilitating infectious diseases (brucellosis, Q fever, toxoplasmosis, and neosporosis). Additionally, we established a correlation profile of various clinical, biochemical, and hormonal parameters against the recorded body weight of each animal. Our study revealed differences in numerous variables of the clinical and biochemical profile among the two tested groups.

## 2. Materials and Methods

### 2.1. Ethical Statement

This study was performed according to standard procedures identified by the Research Board of the Faculty of Veterinary Medicine, South Valley University, Qena, Egypt. The study was approved by the Research Bioethics Committee at South Valley University VM/SVU/23(1)-05. This study was conducted according to ARRIVE guidelines for animal handling and experimentation (https://arriveguidelines.org/arrive-guidelines accessed on 22 December 2022).

### 2.2. Animals of the Study

The current study was conducted on a private Barki sheep flock at Qena governorate, southern Egypt. Barki sheep have been bred from sheep that were raised in the Barca region in eastern Libya, from which the name was derived. These sheep were transported to western Egypt to the Matrouh governorate by the free movement of Bedouins. High numbers of Barki sheep are usually transported from Matrouh to the Beheira governorate because of plentiful agricultural land, animal farms, and also its proximity to Cairo and other big Egyptian cities (Appendix A). Our tested animal flock of Barki sheep was purchased from a private farm in the Beheira governorate in September 2021 at the age of 4 months old with an approximate body weight of 11–13 kg for the purpose of commercial breeding. These animals were female and were kept and fed together from the first day of arrival until 12 months of age. The animals in this study consumed drinking water and a diet composed of dry berseem (*Trifolium alexandrinum* L.) ad libitum and concentrates based on wheat bran, crushed soybean and yellow corn, vitamin premix, and common salt, formulated to meet the proper rations for the growing animals. Qena is located in the southern region of Egypt, which is a hotter and drier place than the Beheira region [14]. During that time, all sheep were subjected to the same conditions of housing, feeding, vaccination, and deworming regimens. All sheep in this experiment received a primary (at 4 months old) and a booster dose (at 5.5 months old) of Covexin^®^ 8 vaccine according to the manufacturer’s protocols (Merck, Rahway, NJ, USA), which is protective against infection with *Clostridium chauvoei, C. septicum, C. novyi* type B, *C. haemolyticum* (known also as *C. novyi* type D), *C. tetani,* or *C. perfringens* types C and D. In addition, the sheep flock was treated with Trimec according to the manufacturer’s instructions (Pharma SWEDE, 10th of Ramadan city, Sharkia, Egypt), which is a broad spectrum anti-parasitic oral suspension (1 mL contains triclabendazole 50 mg and ivermectin 1 mg), at 7 and 10 months old at two drenching times, each 21 days apart to combat the prepatent stages. Sanitary measures were applied to the farm for routine cleanliness and disinfection by glutaraldehyde. At the age of 12 months, despite exposure to similar conditions, a remarkable difference in body weight was observed in some animals (herein referred as stunted/test animals, *n* = 10) compared to others (referred as good body condition/control animals, *n* = 9) (Figure 1). All the investigated sheep showed no obvious differences in appetite and digestive behaviors.

### 2.3. Case History and Clinical Examination

A specified questionnaire was designed to record the information and complaints of the animal owner, the exhibited clinical signs, and management practices. The collected data were organized with our recorded signs and observations on each animal and breeding site. Careful routine clinical examination of these studied cases was carried out according to the methods described previously [15].

### 2.4. Fecal Collection and Parasitological Investigations

Fecal samples were collected directly from the rectum of stunted and good body condition animals in a clean, dry plastic bag for examination of specific eggs using direct smear and a concentration sedimentation-saturated salt-based flotation technique [16].

### 2.5. Blood Collection

At the end of study, 5 mL blood samples were collected via jugular venipuncture from each animal at 12 months old after disinfecting the puncture area with ethyl alcohol 70%. The samples were placed into a vacutainer tube without anticoagulant for serum separation from stunted sheep (*n* = 10) and those of good body condition (*n* = 9). Serum was separated by leaving the sample in an upright position for 2 h in a cooling box followed by centrifugation at 4000 rpm/20 min. Next, the separated serum samples were stored at −20 °C until biochemical analyses.

### 2.6. Biochemical Investigation

Serum samples were used to estimate total protein, albumin, urea, urea nitrogen, creatinine, cholesterol, triglycerides, total bilirubin, direct bilirubin, alanine transaminase (ALT), aspartate transaminase (AST), and glucose using colorimetric methods and commercial kits (Spectrum Diagnostics, Cairo, Egypt) according to the manufacturer’s instructions. Calcium, phosphorus, magnesium, iron, and zinc were measured using colorimetric methods and commercial kits (Bio-Diagnostic, Giza, Egypt) according to the manufacturer’s instructions. Values were read using a UV spectrophotometer (SEAC, Slim, Florence, Italy). Globulin concentration was calculated by subtracting albumin from the corresponding total protein value.

### 2.7. Hormonal Analysis

Quantitative determinations of sheep thyroid stimulating hormone (TSH) and sheep growth hormone (GH) were estimated in undiluted serum samples using an ELISA Kit (MyBiosource, Mumbai, Maharashatra, India). The detection assay procedures and calculations were undertaken according to the manufacturer’s guidelines and protocols.

### 2.8. Serological Investigations and ELISAs

To detect antibodies of *T. gondii*, the serum samples were analyzed with an indirect multi-species ELISA for toxoplasmosis (ID.vet, Grabels, France) according to the manufacturer’s instructions. Serum samples and controls were diluted 1:10. The optical density (OD) obtained was used to calculate the percentage of sample (S) to positive (P) ratio (S/P%) for each of the test samples according to the following formula: S/P (%) = (OD sample − OD negative control)/(OD positive control − OD negative control) × 100. Samples with an S/P% of less than 40% were considered negative; if the S/P% was between 40% and 50%, the result was considered doubtful, and considered positive if the S/P% was greater than 50%.

For antibodies of *N. caninum*, serum samples were analyzed with a competitive multi-species ELISA for neosporosis (ID.vet, Grabels, France). Serum samples and controls were diluted 1:2. The ODs obtained were used to calculate the percentage of sample (S) to negative (N) ratio (S/N%) for each of the test samples according to the following formula: S/N (%) = OD sample/OD negative control × 100. Samples with an S/N% greater than 60% were considered negative; if the S/N% was between 50% and 60% the result was considered doubtful and was considered positive if the S/N% was less than 50%.

Concerning *Brucella* spp., serum samples were analyzed with an indirect multi-species ELISA for brucellosis (ID.vet, Grabels, France). Serum samples and controls were diluted 1:20. The ODs obtained were used to calculate the percentage of sample (S) to positive (P) ratio (S/P%) for each of the test samples according to the following formula: S/P (%) = (OD sample − OD negative control)/(OD positive control − OD negative control) × 100. Samples with an S/P% of less than 110% were considered negative; if the S/P% was between 110% and 120% the result was considered doubtful and was considered positive if the S/P% was greater than 120%.

Similarly, in *Coxiella burnetii* (Q fever) antibody detection, serum samples were analyzed with an indirect multi-species ELISA for Q fever (ID.vet, Grabels, France). Serum samples and controls were diluted 1:50. The ODs obtained were used to calculate the percentage of sample (S) to positive (P) ratio (S/P%) for each of the test samples according to the following formula: S/P (%) = (OD sample − OD negative control)/(OD positive control − OD negative control) × 100. Samples with an S/P% of less than 40% were considered negative; if the S/P% was between 40% and 50% the result was considered doubtful, from 50% to 80% were regarded as positive, and it was considered strong positive if the S/P% was greater than 80%.

The ODs of all ELISA results were read at 450 nm and measured with an Infinite^®^ F50/Robotic ELISA reader (Tecan Group Ltd., Männedorf, Switzerland). A summary of commercial kit ELISAs details is described in Table 1.

### 2.9. Statistical Analyses

Statistical analyses of clinical and biochemical variables between lean and good condition groups were measured by unpaired Student’s *t-* test using GraphPad Prism version 5 (GraphPad Software Inc., La Jolla, CA, USA). The 95% confidence intervals of a proportion, including continuity correction, were calculated using www.vassarstats.net (accessed on 22 December 2022). The results were considered significant when the *p*-value was <0.05 or highly significant when the *p*-value was <0.0001.

### 2.10. Pearson’s Correlation Coefficient

The correlation between the values of body weight and those against body condition score, and a number of biochemical parameters that showed significant differences and growth hormone between the 2 groups, were analyzed using Pearson’s correlation coefficient test. Correlation coefficients were calculated using Pearson’s correlation coefficient: |r| = 0.70, strong correlation; 0.5 < |r| < 0.7, moderately strong correlation; and |r| = 0.3–0.5 weak-to-moderate correlation [17].

## 3. Results

In this study, a BCS scale consisting of four grades was evaluated (in the poor condition or stunted sheep group, 1 is emaciated and 2 is thin; in the good body condition or control sheep group, 3 is moderate and 4 is good body condition). There is a highly significant (*p* ≤ 0.0001) difference between the poor body condition test group and the group with good body conditions in terms of body weight and BCS (Table 2) (Figure 2). Regarding clinical systemic parameters, there was a highly significant increase (*p* ≤ 0.0001) in pulse rate (beat/min) in poor body condition animals (87.1 ± 8.9) compared with the control group (68.6 ± 5.3). Similarly, but with a lower degree of significance, there was a significant increase (*p* ≤ 0.0134) in respiratory rate (breath/min) in stunted animals (30.1 ± 4.3) compared with the control group (24.6 ± 4.5). However, the body temperature in both groups did not change markedly (*p* = 0.203). Concerning other items of clinical investigation, no obvious changes were observed between the test and control groups, including characteristics of digestive, respiratory, nervous, or urogenital systems, or skin, mucous membrane, or lymph nodes (Table 2) (Figure 2). These results suggest the minimal effect of stunted growth on clinical pictures of tested sheep.

The results (Figure 3) (Appendix A) show the serum biochemical variables between the stunted and control groups. These results revealed the significant decrease (*p* ≤ 0.05) in cholesterol (35.7 ± 5.8 vs. 49.8 ± 8.8), triglycerides (36 ± 3.3 vs. 46.9 ± 10.4), creatinine (0.58 ± 0.09 vs. 0.81 ± 0.1), total protein (6.7 ± 0.44 vs. 7.2 ± 0.48), and albumin (3.6 ± 0.27 vs. 4.1 ± 0.4) for stunted vs. good body condition groups, respectively. Oppositely, a marked increase (*p* ≤ 0.05) was observed in stunted sheep compared to the good body condition group in the serum urea level (29.6 ± 3.2 vs. 26.9 ± 2.9) and BUN (13.8 ± 1.5 vs. 12.3 ± 1.4), respectively. However, non-statistically significant variations (*p* ≥ 0.05) were obtained liver function tests (total bilirubin, direct and indirect bilirubin, ALT, AST), globulin, and glucose levels. These results indicate the suffering of stunted sheep from nutritional deficiency and a degree of renal insufficiency.

Similarly, non-statistically significant variations (*p* ≥ 0.05) were observed when a number of macro- and micro-elements were tested (calcium, phosphorus, magnesium, zinc, and iron) (Figure 4) (Appendix A).

Although we observed differences between stunted and control groups concerning the thyroid and growth hormones, they were not statistically significant (*p* > 0.05) (Figure 5) (Appendix A). These results demonstrate the balanced ratio of minerals between the two sheep groups.

Pearson’s correlation coefficient was used to investigate the association between body weights with values of BCS and biochemical parameters and showed significant differences between the two groups. A better correlation was identified for BCS (strong; Pearson r = 0.917) and albumin (moderately strong; Pearson r = 0.683), followed by triglycerides (moderately strong; Pearson r = 0.537), cholesterol (weak-to-moderate; Pearson r = 0.489), growth hormone (weak-to-moderate; Pearson r = 0.4005), and eventually total protein (weak-to-moderate; Pearson r = 0.393) (Figure 6).

Additionally, we estimated the possibility of growth retardation in the studied animals because of debilitating infections. We tested for internal parasites using a direct fecal smear and a concentration sedimentation/flotation technique. Different kinds of ELISAs were used to reveal debilitating protozoan (*T. gondii* and *N. caninum*) or bacterial (*Brucella* spp., and *C. burnetii*) infections that are endemic in our locality. There was no evidence of infection of tested pathogens, except for *T. gondii*, in which only one and two cases were identified as seropositive in test and control groups, respectively (Table 3) (Figure 7).

## 4. Discussion

The most common sheep breeds in Egypt are Ossimi, Rahmani, Saidi, and Barki sheep. The latter has a superior quality of meat and is well adapted to the challenging desert environment, characterized by food shortage high temperature fluctuations. Growth performance of Barki sheep has an important economic value in terms of minimizing the shortage of mutton meat in Egypt. Furthermore, milk production is of great importance for feeding newborn lambs [18].

In our comparative study to evaluate health hazards, systemic clinical examinations were applied and revealed distinct clinical symptoms among sheep of poor and good growth. Our systemic clinical examination revealed higher respiratory and pulse rates in stunted sheep than in control sheep. However, this variation might be attributable to the smaller size of the stunted sheep, as can be observed from body and BCS findings [15]. Most of the other examined clinical parameters were within the normal limits. However, the same stunted sheep also showed rough and rusty body coats, revealing the growth disorders in this group of sheep.

Serum cholesterol and triglycerides represent the most important parameters when a lipid profile is studied; both were decreased in stunted sheep, which showed poor growth rates compared with the control group in a significant manner. This was consistent with Singh et al. (2018) [13], who also reported a lower level of such lipid metabolites in slow growth sheep compared with rapid growth sheep, although it was significant only in the case of triglycerides. In addition, the triglycerides contribute in female puberty to the preparatory stage of conception and gestation [19].

The serum urea and urea nitrogen concentrations were increased significantly in the stunted group compared to the good body condition group. Previous studies also demonstrated that urea levels can be increased during stages of increased nutritional demands. This increase is due to the ability of ruminants to recycle endogenous urea partly into the forestomach to synthesize microbial protein [20]. This compensatory mechanism took place in cases of decreased protein intake or imbalanced endogenous protein homeostasis. This can be achieved via a reduction in urea excretion by the kidneys to maintain a high entry of urea to the rumen. The conservation of urea by the kidney occurs through a decrease in renal plasma flow and glomerular filtration rate, causing a reduction in the filtration of urea by glomeruli and an increase in the reabsorption of urea from tubules to blood [21]. On the contrary, the serum creatinine level was reduced significantly in sheep with a poor growth rate compared with control sheep. This result agrees with the study of Piccione et al. (2009) [22], which determined that the amount of formed creatinine depends on the content of creatinine in the body, which is related to muscle mass, food intake, and synthesis rate of creatinine.

Our results demonstrated that the serum total protein and albumin were decreased significantly in sheep with poor growth rather than good growth sheep. These observations agree with the results of Piccione et al. (2009) [22], who attributed the results to low intake of protein and dehydration.

Unexpectedly, our results revealed that all tested parameters of liver function and glucose level remained within the physiological levels in the investigated groups. Similarly, our tested macro-elements, such as calcium, phosphorus, and magnesium, and micro-elements, such as iron and zinc levels, did show statistical differences between stunted and good growth sheep, although lower levels were recorded in the former group in the case of magnesium and iron. Minerals are required by animals to grow and reproduce. They play important roles as structural components of enzymes and regulate metabolic reactions in the animal body [23]. Normal mineral levels in the serum of all tested sheep are indicative of the normal food supply and food intake, which is also supported by our findings of normal appetite [6].

Lower levels of growth and thyroid hormones were observed in stunted sheep than in good growth sheep. This difference was apparent but not statistically recorded; other observations and previous literatures confirmed the hormonal role in growth disorders in stunted groups. In a previous study, somatotropin administration enhanced puberty in Rahmani ewe lambs. This is due to increased provision of trophic signals (represented by increased Serum IGF-1 secretions) and/or blood-borne metabolites (glucose, cholesterol, and lipids) [7]. Another study reported that melatonin implantation improved the sheep growth and development via increasing the growth hormone release [24]. In the same context, appropriate thyroid gland function and activity of thyroid hormones (TH) are considered crucial to sustain productive performance in domestic animals (growth, milk, hair fiber production), and circulating TH can be considered as an indicator of the metabolic and nutritional status of the animals [25,26].

In an attempt to recognize the reasons behind the difference in growth rate between the test and control groups, we also estimated the subclinical infections of a group of endemic infectious diseases affecting sheep. There was no strong evidence of involvement of infections of *T. gondii*, *N. caninum*, *Brucella* spp., or *C. burnetii*, internal or external parasitic infections. As a result of close observations, when we noticed a difference in the body weight in sheep, infection with endoparasites was the first suspicion. Thus, our first procedure for investigation of the reason for weakness in some sheep was fecal analysis for the detection of different parasitic stages. However, none of the tested sheep were positive for any type of parasites, including coccidian oocysts. This finding might be attributable to the parasitological methods used (screening methods) and the sanitary measures applied to animals (deworming by anthelmintics) and farms (routine cleanliness and disinfection by glutaraldehyde and using clean food).

One limitation for this study is the lack of genetic line/pedigree information for the sheep and data on the relationship/degree of relationship between these sheep. Further studies are required to confirm our hypothesis and the obtained results, particularly using animals with a well-known genetic background and close pedigree.

## 5. Conclusions

Our randomized controlled clinical trial revealed valuable information on detecting biomarkers for growth performance in Barki sheep. This information is crucial to address economic and financial concerns as it is useful in excluding animals with poor growth and development from the livestock industry. Our study revealed that in most cases, the stunted Barki sheep exhibited a normal clinical picture, including appetite, and a normal biochemical profile for minerals and liver function markers. Additionally, the serological testing negated the suffering of such animals from brucellosis, Q fever, toxoplasmosis, and neosporosis. However, levels of nutritionally relevant biomarkers, such as total protein, albumin, cholesterol, and triglycerides, were markedly decreased in stunted sheep compared with normal sheep. These findings were also confirmed by lower levels of growth and thyroid hormones in stunted sheep than in the control sheep. Additional studies are required to investigate the productive and reproductive capacities of Barki sheep to validate the usefulness of this breed in nationwide breeding and farming.

## Figures and Tables

**Figure 1 animals-13-00962-f001:**
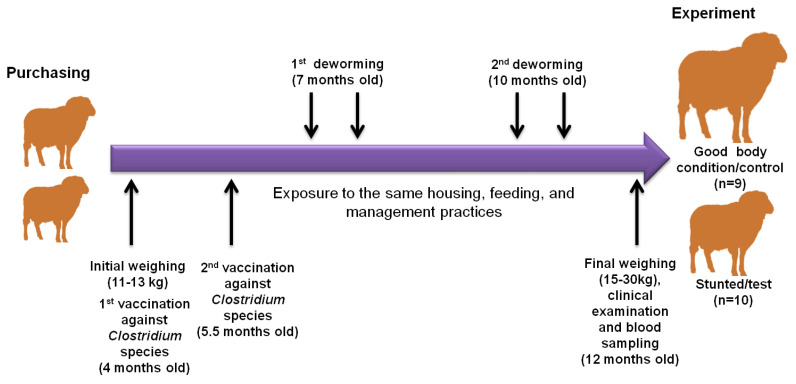
Illustration diagram showing the observational and experimental approaches in this study. Our sheep were purchased from the Beheira governorate in September 2021 at the age of 4 months with an approximate body weight of 11–13 kg. They were subjected to experimental procedures at 12 months old. During that time, all sheep were subjected to the same conditions of housing, feeding, vaccination, and deworming regimens. All sheep in this experiment received a primary (at 4 months old) and a booster dose (at 5.5 months old), and were treated with a broad spectrum anti-parasitic oral suspension at 7 and 10 months old. At the age of 12 months, despite expose to similar conditions, a remarkable difference in body weight was observed in some animals (herein referred as stunted/test animals, *n* = 10) compared to others (referred as good condition/control animals, *n* = 9). The 2 arrows above refer to the drenching of animals with anthelmintic drugs 2 times at 3-week intervals, according to the manufacturer’s instructions.

**Figure 2 animals-13-00962-f002:**
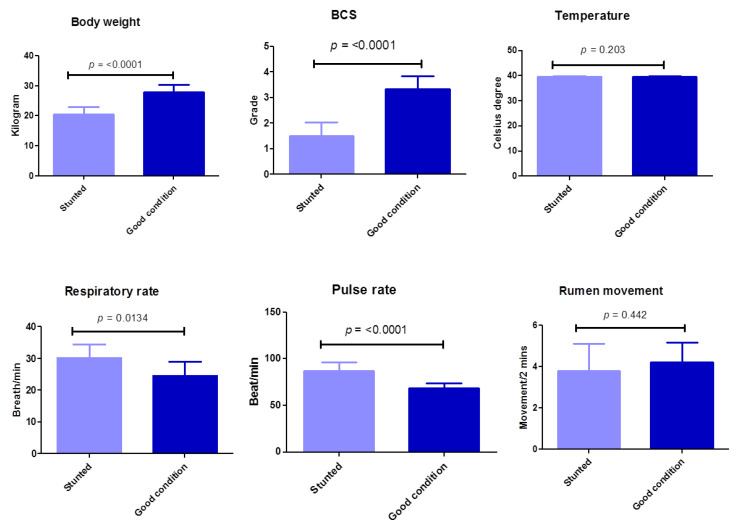
Clinical findings of examined Barki sheep. Body weight, body condition, and systemic parameters were investigated in stunted and good body condition sheep. Each bar represents the mean ± SD with the *p*-value comparing the two groups using an unpaired Student’s *t*-test. Significance at *p* ≤ 0.05.

**Figure 3 animals-13-00962-f003:**
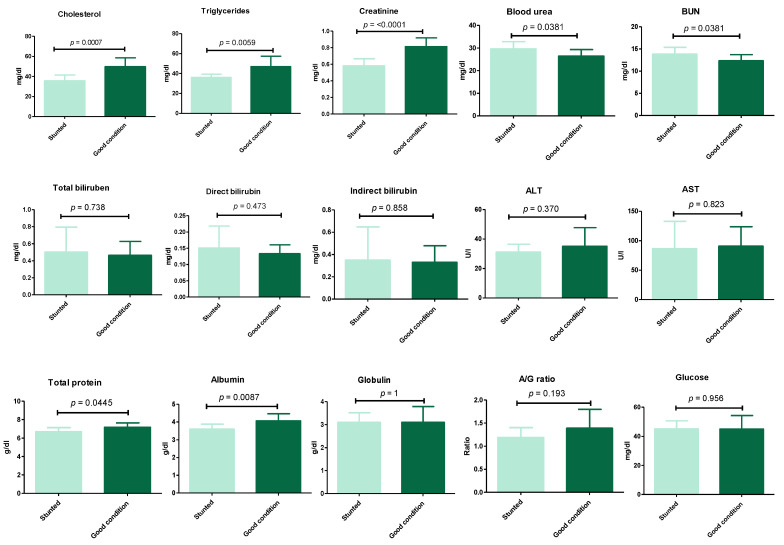
Biochemical analysis of Barki sheep serum. Values of various serum biochemical variables in Barki sheep of stunted and good body condition. Each bar represents the mean ± SD with *p*-value comparing the two groups using an unpaired Student’s *t*-test. Significance at *p* ≤ 0.05.

**Figure 4 animals-13-00962-f004:**
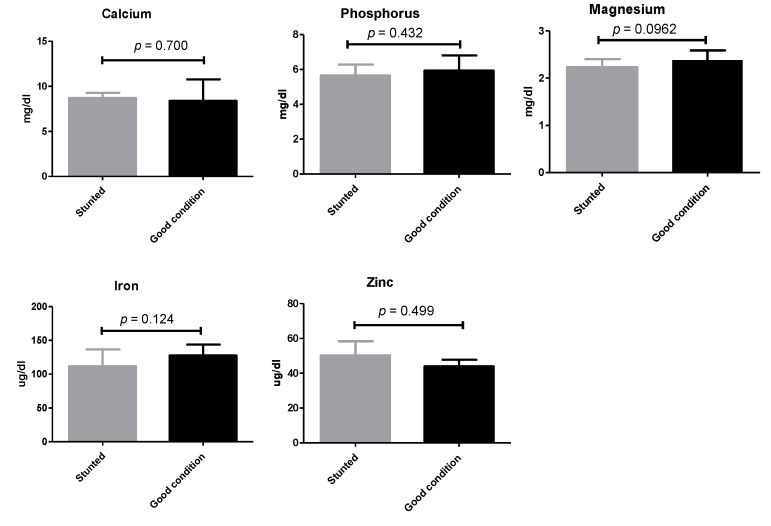
Mineral analysis of Barki sheep serum. Values of various serum variables in Barki sheep of stunted and good body condition. Each bar represents the mean ± SD with *p*-value comparing the two groups using an unpaired Student’s *t*-test. Significance at *p* ≤ 0.05.

**Figure 5 animals-13-00962-f005:**
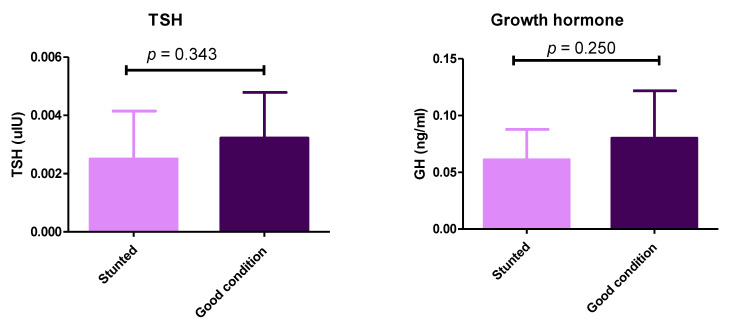
Serum levels of thyroid-stimulating hormone and growth hormone in stunted and good body condition Barki sheep. Serum level of TSH and growth hormone between stunted Barki sheep (*n* = 10) and good body condition sheep or control group (*n* = 9). Each bar represents the mean ± SD with *p*-value comparing the two groups using an unpaired Student’s *t*-test. Significance at *p* ≤ 0.05.

**Figure 6 animals-13-00962-f006:**
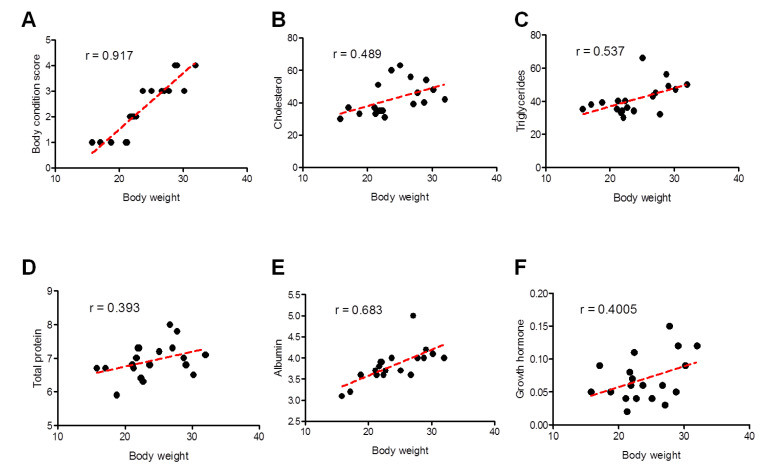
Correlation between different clinical and biochemical parameters against the body weight of all examined sheep. Scatter graphs show the correlation between body weight and different parameters including body condition score (**A**), cholesterol (**B**), triglycerides (**C**), total proteins (**D**), albumin (**E**), and growth hormone (**F**) from stunted (*n* = 10) and good body condition sheep (*n* = 9). The equation represents the approximation formula. The break line represents the calculated line of best fit. Correlation coefficients were calculated using Pearson’s correlation coefficient: |r| = 0.70 strong correlation; 0.5 < |r| < 0.7 moderately strong correlation; and |r| = 0.3–0.5 weak-to-moderate correlation.

**Figure 7 animals-13-00962-f007:**
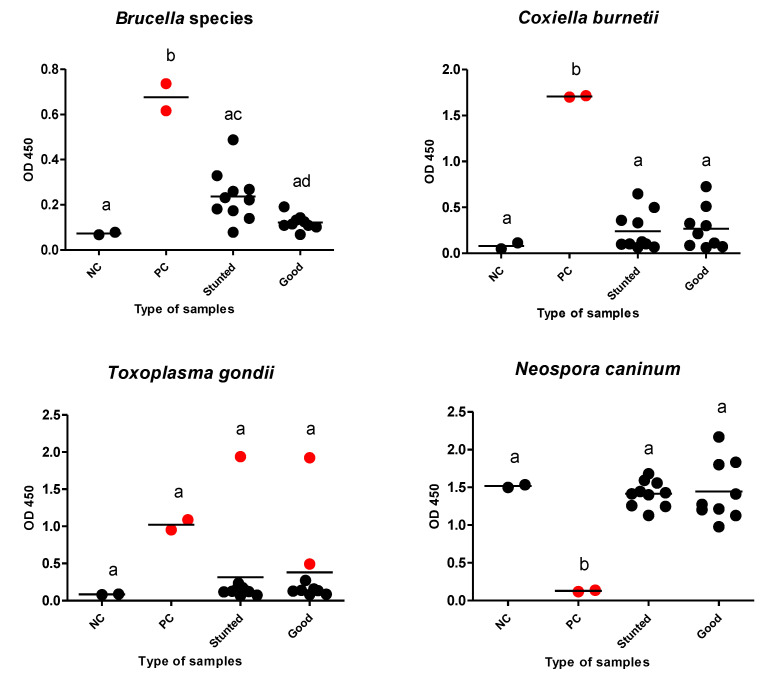
Different ELISAs for examined Barki sheep. Sera from stunted (*n* = 10) and good body condition (*n* = 9) Barki sheep were tested using a commercial ELISA for detection of specific antibodies against *Brucella* species, *Coxiella burnetii*, *Toxoplasma gondii*, and *Neospora caninum* and for significant differences in different sera against the same tested antigen. Each bar represents the mean ± standard deviation. The different letters above the bars in the graphs indicate statistically significant differences of other groups (one-way ANOVA with Tukey–Kramer post hoc analysis, *p* < 0.05). Red dots refer to positive samples either from the control or test samples. NC; negative control, PC; positive control.

**Table 1 animals-13-00962-t001:** Commercially available ELISA test kits used for detecting antibodies to *T. gondii*, *N. caninum*, *Brucella* spp., and Q fever.

Infectious Agent	ELISA Test Kit	Antigen	Serum Dilution	Conjugate	Interpretation
*Toxoplasma gondii*	ID Screen^®^ Toxoplasmosis Indirect Multi-species	P30 antigen	1:10	Anti-multi-species IgG-HRP	Negative < 40 Doubtful 40–50 Positive > 50
*Neospora caninum*	ID Screen^®^ *Neospora caninum* competition Multispecies	Purified extract of *Neospora caninum*	1:2	Anti-*N. caninum*-HRP(detect IgG or IgM)	Negative > 60 Doubtful 50–60 Positive < 50
*Brucella* species	ID Screen^®^ Brucellosis Serum Indirect Multispecies	LPS of *Brucella* species	1:20	Anti-multi-species- IgG-HRP	Negative < 110 Doubtful 110–120 Positive > 120
*Coxiella burnetii* (Q fever)	ID Screen^®^ Q fever Indirect Multispecies	Phase I and II proteins of *C. burnetii*	1:50	Anti-multi-species IgG-HRP conjugate	Negative < 40 Doubtful 40–50 Positive > 50

All kits were purchased from ID. Vet., Innovative Diagnostics, Grabels, France.

**Table 2 animals-13-00962-t002:** Clinical findings among Barki sheep of good and poor condition.

Parameter (Unit)	Stunted/Test (*n* = 10)	Good Condition/Control (*n* = 9)	*p* Value
Body weight (Kg)	20.5 ± 2.4	27.8 ± 2.5	<0.0001
Body condition score (grade 1–4)	1.5 ± 0.52	3.3 ± 0.51	<0.0001
Body temperature (Celsius degree)	39.6 ± 0.23	39.5 ± 0.25	0.203
Respiratory rate (breath/min)	30.1 ± 4.3	24.6 ± 4.5	0.0134
Pulse rate (**beat**/min)	87.1 ± 8.9	68.6 ± 5.3	<0.0001
Ruminal movement (contraction/2 min)	3.8 ± 1.3	4.2 ± 0.97	0.442
Appetite	Normal to slightly reduced	Normal	-
Feces	Normal	Normal	-
Nasal discharges	Scanty to absent	Scanty to absent	-
Coat	Rough, rusty, alopecia	Shiny, smooth	-
Mucous membrane	Pale to rosy-red	Rosy-red	-
Lymph node abnormalities	None	None	-
Eyes lesions	None	None	-
Urogenital abnormalities	None	None	-
Nervous disorders	None	None	-
Locomotor disorders	None	None	-

Significance values at <0.05 *p* value and highly significant at <0.0001 *p* values measured by *t*-test (unpaired).

**Table 3 animals-13-00962-t003:** Serological testing of Barki sheep in good and poor condition.

Parameter	Used Method	Stunted (*n* = 10)	Good Condition (*n* = 9)
*Brucella* spp. antibodies	Indirect ELISA	0	0
** *Coxiella burnetii* ** **antibodies**	Indirect ELISA	0	0
*Toxoplasma gondii* antibodies	Indirect ELISA	1 (10%)	2 (22.2%)
*Neospora caninum* antibodies	Competitive ELISA	0	0

## Data Availability

All data generated and analyzed during this study are included in this published article. Raw data supporting the findings of this study are available from the corresponding author upon request.

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
