# Peer review of "Evaluation of Clinical and Biochemical Traits in Egyptian Barki Sheep with Different Growth Performances"

_animals, 2023, doi:10.3390/ani13060962_

Round 1
Reviewer 1 Report (Previous Reviewer 4)
The authors have made a fair attempt in revising the manuscript. Following are my comments:
Major comments:
1. The authors’ justification for the sheep to not have close relation is hard to accept. Though the Bakri sheep may give birth to one lamb per birth, it is a known fact that usually a single ram (male) is used for breeding a group of ewes (females). So, in this case there is every possibility that the selected lambs could be half sibs too. A half sib relation is also a close relation. Additionally, the fact that all these lambs were selected from a single farm, further increases the possibility for the lambs to be closely related. This therefore raises a big question on the experimental design. Additionally, I urge the authors to delete the statement between lines 475-477 mentioning that the lambs are not closely related.
2. Regarding my first query of the initial review report on the acclimatization [In the abstract the authors have mentioned that the study was conducted on 4 month old sheep. However, in the material and methods they have also stated that the sheep were purchased at 4 months age and were acclimatized for 8 months and the final recording was done at 12 months age. Therefore, there arises a concern as to what exactly was the study period? Kindly clarify this.]. I appreciate the efforts made to address this however following are my concerns:
a. Firstly, acclimatization is the period given for the animals to adjust to the new environment before the beginning of an experiment. So as per the authors’ statement, if the animals were purchased at 4 months age and acclimatized for 8 months, when were they subjected to the experiment? Technically 4 months age plus 8 months acclimatization itself completes 12 months age.
b. Hence, I suppose there seems to be a confusion in the terms used. Firstly, since the authors did not subject the animals to any specific ‘treatment’, its better to avoid the term ‘experiment’. Secondly either avoid using the term ‘acclimatization’ or mention a proper duration of acclimation. Kindly note, if mentioning the acclimation duration for 8 months then technically the observations made during the acclimatization period cannot be considered for the study.
3. What were the total number of lambs purchased for this study? Was it 19 (10 +9).
a. If yes, then could the authors explain if there were any specific reason for the purchase of that many number of animals?
b. If no: kindly mention the total animals purchased. Based on this maybe the study could be improvised so as to assess the proportion of dwarf animals in the lot and thereby link for a possible issue.
4. The authors have stated in the manuscript that the study revealed useful markers for monitoring the growth performance in Egyptian Barki sheep. These findings would have been novel if the markers could be detected at an earlier stage rather than on completion of 12 months. Because this could aid in earlier selection and appropriate culling of less producing/dwarf sheep.
Minor comments:
1. Line 280-282: kindly edit the unit of measurement of pulse rate to beat/min (from beat/1min). Also make the appropriate edits in the graphs for Respiration rate and Pulse rate
Author Response
Comments and Suggestions for Authors
The authors have made a fair attempt in revising the manuscript. Following are my comments:
Author’s response
We greatly appreciate reviewer 1 for the continuous help by giving valuable comments and suggestions for improvement of our study. In below are our point by point answers and responses based on reviewer comments. Changes and revisions in manuscript in the current version are marked by highlighting in bluish coloration.
Major comments:
- The authors’ justification for the sheep to not have close relation is hard to accept. Though the Bakri sheep may give birth to one lamb per birth, it is a known fact that usually a single ram (male) is used for breeding a group of ewes (females). So, in this case there is every possibility that the selected lambs could be half sibs too. A half sib relation is also a close relation. Additionally, the fact that all these lambs were selected from a single farm, further increases the possibility for the lambs to be closely related. This therefore raises a big question on the experimental design. Additionally, I urge the authors to delete the statement between lines 475-477 mentioning that the lambs are not closely related.
Author’s response
We thank reviewer for this valuable comment and also understand the worries behind this point. However, as mentioned in our manuscript in simple summary and in the text and also in our previous version of revision note, this is a randomized controlled clinical study. In such similar studies, it is very difficult to control and standardize all factors and aspects as occurred in experimental studies. Nevertheless, in this study, we attempted to reach the outmost possible approaches of our study standardization. This was performed via selecting animals from the same breed (Barki), sex (female), age (4 months old), and management practice (same farm). Then, such animals were monitored thoroughly by ourselves and another group of well trained persons concerning feeding, housing and receiving vaccination and deworming procedures. Such animals (n = 19) and another group was initially purchased for commercial purposes. During our close observations, marked difference was observed among a group of Barki in spite of receiving the same feed, management and care. Thus, this study was conducted seeking the discovery of reasons and impacts of such problem. Several approaches were performed including various clinical findings and biochemical parameters. Also, Pearson correlation coefficient was tested among variables with recorded statistical significant changes. Although these many novel and significant findings of the current study, we also highlighted the limitation of using animals of unknown genetic pedigree and recommended further studies for confirmation of our results and hypothesis.
We deleted the statement between lines 475-477 based on the reviewer comment.
- Regarding my first query of the initial review report on the acclimatization [In the abstract the authors have mentioned that the study was conducted on 4 month old sheep. However, in the material and methods they have also stated that the sheep were purchased at 4 months age and were acclimatized for 8 months and the final recording was done at 12 months age. Therefore, there arises a concern as to what exactly was the study period? Kindly clarify this.]. I appreciate the efforts made to address this however following are my concerns:
- Firstly, acclimatization is the period given for the animals to adjust to the new environment before the beginning of an experiment. So as per the authors’ statement, if the animals were purchased at 4 months age and acclimatized for 8 months, when were they subjected to the experiment? Technically 4 months age plus 8 months acclimatization itself completes 12 months age.
- Hence, I suppose there seems to be a confusion in the terms used. Firstly, since the authors did not subject the animals to any specific ‘treatment’, its better to avoid the term ‘experiment’. Secondly either avoid using the term ‘acclimatization’ or mention a proper duration of acclimation. Kindly note, if mentioning the acclimation duration for 8 months then technically the observations made during the acclimatization period cannot be considered for the study.
Author’s response
We greatly thank reviewer for this important point. At first, we included the time of acclimatization 8 months (starting from purchasing time 4 months until termination of study at 12 months old) because this period included vaccination and deworming procedures. However, we totally agree with the reviewer comment and we deleted such term of acclimatization and all relevant sentences considering 12 months old sheep as exact time of experimentation and previous period for close observation and routine management as shown in figure 1. All relevant changes were highlighted in blue colour in the manuscript.
- What were the total number of lambs purchased for this study? Was it 19 (10 +9).
- If yes, then could the authors explain if there were any specific reason for the purchase of that many number of animals?
- If no: kindly mention the total animals purchased. Based on this maybe the study could be improvised so as to assess the proportion of dwarf animals in the lot and thereby link for a possible issue.
Author’s response
Actually as we mentioned before, this farm contains higher number than those specified for this study. The farm contains 40 Barki sheep of similar breed, age, sex as mentioned earlier that have been purchased for commercial and research purposes. During our close supervision of this farm, we noticed that 10 sheep showed a marked retarded growth comparing to others (n = 30) that housed at the same place, feed on the same foods, and exposed to the same management practices. As a control, we chose another group of sheep of similar age and sex that housed together and feed on the same foods (n = 9 from 30). We could not use high higher number for using as control because we deemed that 9 animals are sufficient and also for financial issues.
Also, the aim of this study was to evaluate the alterations in some clinical and biochemical traits in such group of animals but not to assess the proportion of dwarfism because we thought that even 40 animals are also not enough to assess the incidence of dwarfism thus we assessed the health impact and associated alterations instead.
- The authors have stated in the manuscript that the study revealed useful markers for monitoring the growth performance in Egyptian Barki sheep. These findings would have been novel if the markers could be detected at an earlier stage rather than on completion of 12 months. Because this could aid in earlier selection and appropriate culling of less producing/dwarf sheep.
Author’s response
We agree with the reviewer comment and revised the relevant statement in the manuscript. However, based on this study and many previous studies our recorded markers (albumin, total proteins, triglycerides and growth hormone) are known as useful markers for monitoring or revealing growth or nutritional impairments. In our study, we additionally established correlation graphs among such markers and body weights of the two groups of animals.
Minor comments:
- Line 280-282: kindly edit the unit of measurement of pulse rate to beat/min (from beat/1min). Also make the appropriate edits in the graphs for Respiration rate and Pulse rate
Author’s response
The units were revised in the text and graphs.

Reviewer 2 Report (Previous Reviewer 3)
happy with the revised version
Author Response
We greatly appreciate reviewer 2 for favorable evaluation of our study. based on reviewers and editorial comments we revised some parts in our manuscript as in attached file for our manuscript.

Reviewer 3 Report (Previous Reviewer 1)
Thank you for updating the manuscript based on reviewer comments.
Author Response
We greatly appreciate reviewer 3 for invaluable comments and favorable evaluation of our study. Based on reviewer comments, some revisions had been made in our manuscript as shown in highlighted parts in bluish coloration in the attached manuscript file. Hopefully this revisions will be sufficient for acceptance of study for publication.

Round 2
Reviewer 1 Report (Previous Reviewer 4)
I appreciate the efforts taken by the authors to improvise the manuscript. I very well understand the practical constraint and appreciate the justifications given by the authors. Congratulations to all the authors for the good work.
This manuscript is a resubmission of an earlier submission. The following is a list of the peer review reports and author responses from that submission.
Round 1
Reviewer 1 Report
The purpose of this paper was to evaluate biochemical characteristics in of animals of different growth rates. Overall, the paper is easy to follow and the results make sense. The paper could benefit from a more thorough editing.
Comments:
-There is no simple summary provided.
Line 29-32: No p-values are provided.
Line 36-39: No p-values are provided.
Line 45/46: Introduction is listed twice.
Other comments: In the introduction the paper says multiple breeds of sheep were used, however throughout the manuscript the authors suggest that only the Barki breed was evaluated. As such this will need clarified throughout the manuscript.
Additionally, as the sheep came from different farms it would be interesting to know if all of the stunted sheep came from a similar area or breeder.
Author Response
Responses to reviewer 1 comments
Reviewer comment
The purpose of this paper was to evaluate biochemical characteristics in of animals of different growth rates. Overall, the paper is easy to follow and the results make sense. The paper could benefit from a more thorough editing.
Author’s response
We greatly appreciate reviewer 1 encouraging words and also the valuable comments and suggestions that greatly helped us to improve the quality of our manuscript. Herein and in the manuscript main and supplemental files we revised and modified manuscript based on reviewer comments.
Reviewer comment
-There is no simple summary provided.
Author’s response
We prepared simple summary as follows;
“Our randomized controlled clinical study revealed novel and valuable information on detecting biomarkers for growth performance in Barki sheep. Our study revealed that in most cases the stunted Barki sheep exhibited a normal clinical picture including appetite, and normal biochemical profile for minerals and liver function markers. However, levels of nutritional relevant biomarkers such as total protein, albumin, cholesterol and triglycerides were markedly decreased in stunted than in normal sheep. These findings were also confirmed by lower levels of growth and thyroid hormones in stunted sheep than those in the control sheep. This approach is useful from the economic and financial aspects as it can be applied for excluding animals with poor growth and development from the livestock industry.”
Reviewer comment
Line 29-32: No p-values are provided.
Author’s response
P value was added as “p < 0.05”.
Reviewer comment
Line 36-39: No p-values are provided.
Author’s response
P value was added as “p < 0.05”.
Reviewer comment
Line 45/46: Introduction is listed twice.
Author’s response
Duplicated word was deleted.
Reviewer comment
Other comments: In the introduction the paper says multiple breeds of sheep were used, however throughout the manuscript the authors suggest that only the Barki breed was evaluated. As such this will need clarified throughout the manuscript.
Author’s response
We apologize for this confusion; We meant by this sentence that our study comparing to a previous report [13], that we used a different breed, sex, location. We revised the English of this sentence as follows;
From
“A previous report had investigated some endocrine and biochemical parameters among fast and slow growing lambs at postweaning period [13]. However, in the current study, we investigated sheep of different breeds, sexes, locations, and demonstrated novel data”.
To
“A previous report had investigated some endocrine and biochemical parameters among fast and slow growing Muzaffarnagari male and female lambs at postweaning period [13]. However, in the current study, we investigated sheep of a different breed (Barki), sexe (female), location (Egypt), and demonstrated novel data on….”
Reviewer comment
Additionally, as the sheep came from different farms it would be interesting to know if all of the stunted sheep came from a similar area or breeder.
Author’s response
In this study, all our sheep were purchased from the same private farm at Beheira. Thus, all sheep were subjected to the same housing, feeding and management practices. However, no pedigree for each animal was established to recognize the relative animals.
We revised the following sentence for more clarification
“Our experimental animal flock of Barki sheep was purchased from Beheira governorate in September 2021 at the age of 4 months old”
To
“Our experimental animal flock of Barki sheep was purchased from a private farm at Beheira governorate in September 2021 at the age of 4 months old.
Reviewer 2 Report
Dear editor
in my opinion the manuscript has several gaps and cannot be accepted for publication especially for a special issue related to parasites.
In particular, I contest the authors' organization of the research which does not allow for relevant results or at least clear and evident results. This is highlighted more specifically for the parasitological part. The authors want to verify if there are differences between 2 groups of Barki sheep, a control group consisting of 9 sheep in good condition and a group of stunted animals (10 sheep). The animals resulted not parasitized by endoparasites and in particular by gastrointestinal nematodes which represent the significant problem of small grazing ruminants. This result can be explained by the following reasons: 1) the animals were treated twice at 7 and 10 months of age and samples were taken at 12 months. Although the animals were treated with products that have no residual activity, the time between the last treatment and sampling is compatible with the prepatent period of many of the gastrointestinal nematodes of sheep. However, the administration of two treatments and the keeping of the animals indoors (?) probably prevented the re-infestation of the Barki sheep and it is therefore obvious that a comparison under these conditions is useless. I would also add that they should at least have tested positive for Eimeria infections, all the more so since they are young animals and the products administered do not work on these protozoa. The parasitological methods used by the authors are certainly not very sensitive, but they should have found coccidian oocysts.
Another limit in my opinion, in addition to the young age of the subjects, is the low number that may have invalidated the results both for gastrointestinal nematodes and Toxoplasma. See the data by Ibrahim Abbasa and Michael B.Hildreth (Trichostrongyle infections in domestic ruminants from Egypt: A systematic review and meta-analysis) in which the prevalence of infection of trichostrongyles is equal to 45%. Furthermore, the same authors find a prevalence of 46.1% for Toxoplasma (Seroprevalence of Specific Antibodies to Toxoplasma gondii, Neospora caninum, and Brucella spp. in sheep and goats in Egypt, Ragab M. Fereig et al 2022, Animals, 2022, 12 (23), 3327) whereas they found a prevalence of infection of 15.78% (3/19) in present manuscript.
Furthermore, I wondered how it is possible that animals fed in the same way, with the same type of management can show such evident differences in physical conditions. Did the authors consider all the causes that led to these differences? It is also strange that animals are divided into two categories that have the same number of animals (9 and 10).
The other point that I dispute is related to the purpose of demonstrating that Barki sheep have characteristics that make them more suitable for breeding than other breeds; then the authors should have compared the Barki with a cosmopolitan race.
Among other gaps, I contest among other things statements in the introduction such as the importance of neospore infection in small ruminants which is obviously not as important as in cattle.
There are no aims clearly described, but the introduction ends with an emphasis on the results obtained.
Author Response
Responses to reviewer 2 comments
Author’s response
We greatly appreciate reviewer 2 for the valuable comments and suggestions that greatly helped us to improve the quality of our manuscript. Herein and in the manuscript main and supplemental files we revised and modified manuscript based on reviewer comments.
Reviewer comment
In my opinion the manuscript has several gaps and cannot be accepted for publication especially for a special issue related to parasites.
Author’s response
We thank the reviewer 2 for insightful comments and criticism. The current study is a controlled randomized clinical trial that showed novel data on numerous clinical and biochemical, and hormonal variables associating Barki sheep with different growth rates. In this study, we used a group of sheep that could be examined and monitored closely by ourselves as supervisors for such farm. In the beginning, we selected clinically healthy, females and 4 months old Barki lambs for the purpose of breeding for commercial purpose. During our supervision with considering all standards good feeding, breeding, and management practices, we noticed a remarkable difference in growth rates among the same group of sheep. This difference was observed although all housed sheep were showing normal feeding behavior and not suffering from any clinical illness or diseases. However, in the revised version of manuscript, we performed substantial changes according to all reviewers’ comments and suggestions.
Regarding the point of inappropriateness of publication in special issue related to parasites, we are accepting to change to another special issue according to animals journal editorial office suggestions.
Reviewer comment
In particular, I contest the authors' organization of the research which does not allow for relevant results or at least clear and evident results. This is highlighted more specifically for the parasitological part. The authors want to verify if there are differences between 2 groups of Barki sheep, a control group consisting of 9 sheep in good condition and a group of stunted animals (10 sheep). The animals resulted not parasitized by endoparasites and in particular by gastrointestinal nematodes which represent the significant problem of small grazing ruminants. This result can be explained by the following reasons: 1) the animals were treated twice at 7 and 10 months of age and samples were taken at 12 months. Although the animals were treated with products that have no residual activity, the time between the last treatment and sampling is compatible with the prepatent period of many of the gastrointestinal nematodes of sheep. However, the administration of two treatments and the keeping of the animals indoors (?) probably prevented the re-infestation of the Barki sheep and it is therefore obvious that a comparison under these conditions is useless. I would also add that they should at least have tested positive for Eimeria infections, all the more so since they are young animals and the products administered do not work on these protozoa. The parasitological methods used by the authors are certainly not very sensitive, but they should have found coccidian oocysts.
Author’s response
We thank reviewer 2 for this valuable comment. Actually, as mentioned earlier, when we noticed a difference in the body weight gain in used sheep, infection with endoparasites was the first suspicion. Thus, our first procedure for investigation of the reason of weakness of some sheep was the fecal analysis for detection of different parasitic stages. However, none of the tested sheep were positive for any type of parasites including coccidian oocysts. This finding might be attributable to the used parasitological methods (screening methods) and applied sanitary measures for animals (deworming by anthelmintics) and farms (routine cleanliness and disinfection by glutaraldehyde and using clean food). Similar information was added in the manuscript in materials and methods and discussion.
Reviewer comment
Another limit in my opinion, in addition to the young age of the subjects, is the low number that may have invalidated the results both for gastrointestinal nematodes and Toxoplasma. See the data by Ibrahim Abbasa and Michael B.Hildreth (Trichostrongyle infections in domestic ruminants from Egypt: A systematic review and meta-analysis) in which the prevalence of infection of trichostrongyles is equal to 45%. Furthermore, the same authors find a prevalence of 46.1% for Toxoplasma (Seroprevalence of Specific Antibodies to Toxoplasma gondii, Neospora caninum, and Brucella spp. in sheep and goats in Egypt, Ragab M. Fereig et al 2022, Animals, 2022, 12 (23), 3327) whereas they found a prevalence of infection of 15.78% (3/19) in present manuscript.
Author’s response
We agree that the used number is not appropriate for seroprevalence of parasitic or bacterial infections. However, this was not the main aim of this study, but we sought to investigate the role or the effect of such kind of infections that were common among sheep in our investigated area on the clinical and biochemical characteristics in Barki sheep.
As mentioned earlier, if this study is not suitable for publication in special issue related to parasites, we are accepting the change to another special issue according to animals journal editorial office suggestions.
Reviewer comment
Furthermore, I wondered how it is possible that animals fed in the same way, with the same type of management can show such evident differences in physical conditions. Did the authors consider all the causes that led to these differences? It is also strange that animals are divided into two categories that have the same number of animals (9 and 10).
Author’s response
As we mentioned earlier, this is a randomized controlled clinical trial, in which a group of lambs with similar breed (Barki), sex (female), age (4 months old) and clinical and physical characteristics were chosen primarily for commercial purposes. As our task as supervisors for this farm, we noticed that 10 sheep showed a marked retarded growth comparing to others that housed at the same place, feed on the same foods, and exposed to the same management practices. As a control, we chose another group of sheep of similar age and sex that housed together and feed on the same foods (n = 9). Actually, the farm contains more numbers that can be considered as control or normal growth rate, but we only selected 9 as a control group.
Reviewer comment
The other point that I dispute is related to the purpose of demonstrating that Barki sheep have characteristics that make them more suitable for breeding than other breeds; then the authors should have compared the Barki with a cosmopolitan race.
Author’s response
In this study, we focused on Barki sheep because it has many characteristics comparing to other Egyptian sheep breeds including high quality of meat with very low fat content, good adaptability to harsh environmental conditions and scarcity of feeds. This recommendation was also stated by many previous reports (Ahmed, 2008; Abousoliman et al., 2020).
Ahmed, A.M. 2008. Technical and financial analysis of Barki sheep under semi-intensive production system. Egyptian J. Anim. Prod. 45: 25-34.
Abdel – Moneim, A.Y.; Ahmed A.M.; Ibrahim M.M.; Mokhtar, M.M. 2009. Flock dynamics of desert Barki sheep in relation to age structure. Trop Anim Health Prod (2009) 41:899–905. DOI 10.1007/s11250-008-9277-4
Abousoliman I, Reyer H, Oster M, Muráni E, Mourad M, Rashed MA, Mohamed I, Wimmers K. Analysis of Candidate Genes for Growth and Milk Performance Traits in the Egyptian Barki Sheep. Animals (Basel). 2020 Jan 23;10(2):197. doi: 10.3390/ani10020197.
Reviewer comment
Among other gaps, I contest among other things statements in the introduction such as the importance of neospore infection in small ruminants which is obviously not as important as in cattle.
Author’s response
We partially agree with reviewer 2 comment because N. caninum is a major causative agent of economical losses among cattle farms but recently many reports correlated clinical forms of neosporosis including abortion, foetal anomalies and miscarriage in sheep (Nayeri et al., 2022; Basso et al., 2022; Irehan et al., 2022; Withoeft et al., 2022).
Nayeri T, Sarvi S, Moosazadeh M, Daryani A. The Global Prevalence of Neospora caninum Infection in Sheep and Goats That Had an Abortion and Aborted Fetuses: A Systematic Review and Meta-Analysis. Front Vet Sci. 2022 Apr 26;9:870904. doi: 10.3389/fvets.2022.870904. PMID: 35558895; PMCID: PMC9090472.
Basso W, Holenweger F, Schares G, Müller N, Campero LM, Ardüser F, Moore-Jones G, Frey CF, Zanolari P. Toxoplasma gondii and Neospora caninum infections in sheep and goats in Switzerland: Seroprevalence and occurrence in aborted foetuses. Food Waterborne Parasitol. 2022 Aug 17;28:e00176. doi: 10.1016/j.fawpar.2022.e00176. PMID: 36039091; PMCID: PMC9418186.
Irehan B, Sonmez A, Atalay MM, Ekinci AI, Celik F, Durmus N, Ciftci AT, Simsek S. Investigation of Toxoplasma gondii, Neospora caninum and Tritrichomonas foetus in abortions of cattle, sheep and goats in Turkey: Analysis by real-time PCR, conventional PCR and histopathological methods. Comp Immunol Microbiol Infect Dis. 2022 Oct;89:101867. doi: 10.1016/j.cimid.2022.101867. Epub 2022 Aug 27. PMID: 36087449.
Withoeft JA, Da Costa LS, Marian L, Baumbach LF, Do Canto Olegário J, Miletti LC, Canal CW, Casagrande RA. Microcephaly and hydrocephalus in a sheep fetus infected with Neospora caninum in Southern Brazil - Short communication. Acta Vet Hung. 2022 Sep 21. doi: 10.1556/004.2022.00028. Epub ahead of print. PMID: 36129791.
Reviewer comment
There are no aims clearly described, but the introduction ends with an emphasis on the results obtained.
Author’s response
Aim of the study was clarified in the abstract and introduction as follows;
“Herein, we proposed to unravel the differences in the clinical and biochemical profiles among Barki sheep of different growth rates. We measured clinical and biochemical parameters in stunted (n = 10; test group) and in good body condition (n = 9; control group) Barki sheep.”

Reviewer 3 Report
The authors evaluated clinical and biochemical traits in Barki sheep. While the study seems interesting and novel, several issues need addressing:
The genus and spices names should be italicised throughout the manuscript.
Is there a particular reason why the authors only focussed on Toxoplasma and Neospora? There is no mention of internal parasites and their potential role.
Figure 1: Why did the authors use two arrows for deworming? Was each animal drenched twice on each occasion?
section 2.5. How many times was the blood collected, and how much?
What's the difference between Figure 3 and Table 3? It looks like the same data is presented in two different ways.
Line 380, isn't high pulse and increased respiration common in stunted animals?
The authors need to compare and discuss these findings with the findings, if any, from other Egyptian sheep breeds
Author Response
Responses to reviewer 3 comments
Reviewer comment
The authors evaluated clinical and biochemical traits in Barki sheep. While the study seems interesting and novel, several issues need addressing:
Author’s response
We greatly appreciate reviewer 3 encouraging words and also the valuable comments and suggestions that greatly helped us to improve the quality of our manuscript. Herein and in the manuscript main and supplemental files we revised and modified manuscript based on reviewer comments.
Reviewer comment
The genus and spices names should be italicised throughout the manuscript.
Author’s response
Done throughout the text.
Reviewer comment
Is there a particular reason why the authors only focussed on Toxoplasma and Neospora? There is no mention of internal parasites and their potential role.
Author’s response
Herein, to exclude the debilitating infectious diseases as a cause of emaciation or weakness, we screened some infections such as Brucella, Coxiella burnetii, Toxoplasma gondii, Neospora caninum and internal parasites. We added some information explaining the reasons as follows;
In abstract
“To detect the reasons for emaciation, certain debilitating infections have been tested. All the tested sheep showed negative coprological tests for gastrointestinal parasites, and had no obvious seropositivity to brucellosis, toxoplasmosis, neosporosis or Q fever.”
In introduction
“Also, herein, we provided more evidence for the health status of tested sheep using serological screening of a group of highly debilitating infectious diseases (brucellosis, Q fever, toxoplasmosis, and neosporosis).”
In results
“Additionally, we estimated the possibility of growth retardation in the studied animals because of debilitating infections. We tested for internal parasites using direct fecal smear and concentration sedimentation/flotation technique. Also, different kinds of ELISAs have been used to reveal debilitating protozoan (T. gondii and N. caninum) or bacterial (Brucella spp., and C. burnetii) infections that are endemic in our locality. No evidence of infections of tested pathogens except for T. gondii in which only one and two cases were identified as seropositive in test and control groups, respectively (Table 3) (Figure 7).”
In discussion
“In an attempt to recognize the reasons for difference in growth rate among the test and control groups, we also estimated the subclinical infections of a group of endemic infectious diseases affecting sheep. No strong evidence of involvement of infections to T. gondii, N. caninum, Brucella spp., or C. burnetii, internal or external parasitic infections.”
Reviewer comment
Figure 1: Why did the authors use two arrows for deworming? Was each animal drenched twice on each occasion?
Author’s response
The two arrows refer to drenching of animals by the anthelmintics drugs for 2 times by 3 weeks intervals according to manufacturer instructions. The time between each drenching (21 days) is compatible with the prepatent period of many of the gastrointestinal nematodes of sheep. We added similar information in figure 1 legend and also in materials and methods section as follows;
” In addition, the sheep flock was treated with Trimec according to the manufacturer’s instructions (Pharma SWEDE, 10th of Ramadan city, Sharkia, Egypt), which is a broad spectrum anti-parasitic oral suspension (1 ml contains triclabendazole 50mg and ivermectin 1 mg) at 7 and 10 months old by two drenching times each with 21 days apart to combat the prepatent stages.”
Reviewer comment
section 2.5. How many times was the blood collected, and how much?
Author’s response
We revised this part as follows;
“At the end of study, an amount 5 mL of blood samples was collected via jugular venipuncture from each animal at 12 months old after disinfecting the puncture area with ethyl alcohol 70% into a vacutainer tube.”
Reviewer comment
What's the difference between Figure 3 and Table 3? It looks like the same data is presented in two different ways.
Author’s response
We added Table 3 because it has more data. However, we moved Table 3 to supplementary and renamed as Table S1.
Reviewer comment
Line 380, isn't high pulse and increased respiration common in stunted animals?
The authors need to compare and discuss these findings with the findings, if any, from other Egyptian sheep breeds
Author’s response
Usually this is a reversible relationship among respiration and pulse against body size. Because there is no published literature revealing the effect of poor growth on respiratory and pulse rate, we had explained this point based on a very common text book for veterinary clinical medicine as follows;
“Our systemic clinical examination revealed higher respiratory and pulse rates in stunted sheep than the control ones. However this variation might be attributable to smaller size of such stunted sheep as can be observed from body and BCS findings [15].”

Reviewer 4 Report
The authors have made a good attempt to compile the findings obtained from their study. Below mentioned are some of my concerns and remarks:
1. In the abstract the authors have mentioned that the study was conducted on 4 month old sheep. However, in the material and methods they have also stated that the sheep were purchased at 4 months age and were acclimatized for 8 months and the final recording was done at 12 months age. Therefore, there arises a concern as to what exactly was the study period? Kindly clarify this
2. From what I understand, the grouping of animals into control and stunted groups were made after 12 months age based on the sheep’s performance/appearance. Therefore, there also arises the major question on the genetic line/ pedigree information of these sheep. Does the author have any information on this? How about the relation/ degree or relationship between these sheep? There could be a possibility for some animals being closely related and hence being under the same group? How would the author address this concern?
3. Line 65: Kindly expand the hormones GnRH and LH. Likewise on line 67: IGF-1
4. Line 255 and line 256: kindly indicate the unit of measurement for the variable (pulse rate) while indicating its values. Kindly do the same while mentioning the values of other variables too (respiration rate)
5. Minor language edits and other minor errors:
a. Line 258: …did not markedly altered => did not alter markedly
b. Line 389: Singh et al. (2018) [13] => Singh et al. (2018) [13],
6. Figure 2: Bodyweight: there seems to be a typing error on the y-axis label=> kilogram
7. The usual representation of respiration rate and pulse rate is as breaths/minute and beats per minute, respectively. Hence, I would suggest to the change movement/minute and pulse/minute accordingly.
8. Line 279: kindly delete the brackets ‘()’ either from the table or figure description as I suppose that was a typing error
9. Table 3: kindly expand the abbreviated terms in the footnote under the table. For example: ALT, AST, TSH
10. Table 2- Figure 2; Table 3- Figure 3; Table 3- Figure 4; Table 3- Figure 5: all these indicate the same data (in the table and figure) represented in 2 forms hence it would be better to retain any one of them. That is either representation as table or figure. Having the same data represented both as table and figure (for e.g Table 2 and figure 2 which is the same data represented in 2 forms) does not serve any specific purpose.
11. Line 328-330: minor grammatical error: Pearson’s correlation coefficient was used to investigate the association between body weights and values of BCS and biochemical parameters showed significant… => possible editing: Pearson’s correlation coefficient was used to investigate the association between body weight with BCS and biochemical parameters, that showed significant…
Author Response
Responses to reviewer 4 comments
Reviewer comment
The authors have made a good attempt to compile the findings obtained from their study. Below mentioned are some of my concerns and remarks:
Author’s response
We greatly appreciate reviewer 4 encouraging words and also the valuable comments and suggestions that greatly helped us to improve the quality of our manuscript. Herein and in the manuscript main and supplemental files we revised and modified manuscript based on reviewer comments.
- In the abstract the authors have mentioned that the study was conducted on 4 month old sheep. However, in the material and methods they have also stated that the sheep were purchased at 4 months age and were acclimatized for 8 months and the final recording was done at 12 months age. Therefore, there arises a concern as to what exactly was the study period? Kindly clarify this.
Author’s response
Such animals had been initially purchased at the age of 4 months old at which initial weighing was performed, and monitored for 8 months for feeding, housing, and management practices. While, 12 months old was the time of termination at which blood sampling, weighing, clinical observations and other experimental approaches have been conducted.
These points had been explained in the manuscripts as follows;
In abstract and introduction
“Animals subjected to this experiment were of the same sex (female), age (starting at 4 months old and terminated at 12 months old), and housed in the same farm with similar conditions of feeding, management practice, and vaccination and deworming regimens.”
In materials and methods
“Our experimental animal flock of Barki sheep was purchased from a private farm at Beheira governorate in September 2021 at the age of 4 months old and a proximate body weight of 11-13 kg, for the purpose of commercial breeding. These animals are composed of females and were kept and fed together from the first day of arrival until 12 months of age. Because of the change in geography and climate between the birth place and breeding place, approximately 8 months were allowed for optimal acclimatization of the tested sheep.”
- From what I understand, the grouping of animals into control and stunted groups were made after 12 months age based on the sheep’s performance/appearance. Therefore, there also arises the major question on the genetic line/ pedigree information of these sheep. Does the author have any information on this? How about the relation/ degree or relationship between these sheep? There could be a possibility for some animals being closely related and hence being under the same group? How would the author address this concern?
Author’s response
We thank the reviewer 4 for this valuable comment. Herein, we used Barki sheep at 4 months old as a starting point for this experiment. Such animals were purchased from one farm, from which we confirmed numerous points regarding information on feeding, age, housing and breeding. Nevertheless, the genetic line/ pedigree information of used sheep was not available at this farm. However, Barki sheep is a well-known breed that giving one lamb per birth, and these animals were of the same age (4 months old) suggesting no close relationship among tested sheep.
We referred to this point as a limitation of this study as follows;
“One limitation for this study is lacking the genetic line/ pedigree information of used sheep, and data on the relation/ degree of relationship between these sheep. However, such animals were purchased from the same farm, from which we confirmed numerous points regarding information on feeding, age, housing and breeding. However, Barki sheep is a well-known breed that giving one lamb per birth, and these animals were of the same age (4 months old) suggesting no close relationship among tested sheep.”
- Line 65: Kindly expand the hormones GnRH and LH. Likewise on line 67: IGF-1
Author’s response
Expanded and sentence revised as follows;
“Growth hormone administration led to early puberty in Rahmani ewe lambs this may be due to increased provision of trophic signals represented by increased Gonadotropin-releasing hormone (GnRH) and luteinizing hormone (LH) secretions….”
- Line 255 and line 256: kindly indicate the unit of measurement for the variable (pulse rate) while indicating its values. Kindly do the same while mentioning the values of other variables too (respiration rate)
Author’s response
Added for pulse rate as (beat / 1 min) and for respiratory rate as (breath / 1 min)
- Minor language edits and other minor errors:
- Line 258: …did not markedly altered => did not alter markedly
Author’s response
Corrected. Line ????
- Line 389: Singh et al. (2018) [13] => Singh et al. (2018) [13],
Author’s response
Done. Line ????
- Figure 2: Bodyweight: there seems to be a typing error on the y-axis label=> kilogram
Author’s response
Corrected. Figure 2.
- The usual representation of respiration rate and pulse rate is as breaths/minute and beats per minute, respectively. Hence, I would suggest to the change movement/minute and pulse/minute accordingly.
Author’s response
Revised. Figure 2
- Line 279: kindly delete the brackets ‘()’ either from the table or figure description as I suppose that was a typing error
Author’s response
The sentence was revised from
The present results in (Table 3) (Figure 3), show the serum biochemical variables among the stunted and control groups.
To
The present results (Figure 3) (Table S1), show the serum biochemical variables among the stunted and control groups.
- Table 3: kindly expand the abbreviated terms in the footnote under the table. For example: ALT, AST, TSH
Author’s response
Abbreviated terms were expanded in the footnote. Based on reviewers comment, we moved Table to supplementary files and identified as Table S1.
- Table 2- Figure 2; Table 3- Figure 3; Table 3- Figure 4; Table 3- Figure 5: all these indicate the same data (in the table and figure) represented in 2 forms hence it would be better to retain any one of them. That is either representation as table or figure. Having the same data represented both as table and figure (for e.g Table 2 and figure 2 which is the same data represented in 2 forms) does not serve any specific purpose.
Author’s response
Based on reviewers comment, we moved Table 3 to supplementary files and identified as Table S1.
- Line 328-330: minor grammatical error: Pearson’s correlation coefficient was used to investigate the association between body weights and values of BCS and biochemical parameters showed significant… => possible editing: Pearson’s correlation coefficient was used to investigate the association between body weight with BCSand biochemical parameters, that showed significant…
Author’s response
We apologize for this mistake; sentence was revised as follows;
“Pearson’s correlation coefficient was used to investigate the association between body weights with values of BCS and biochemical parameters, that showed significant difference among the two groups.”
